# Functional Conservation and Divergence of *MOS1* That Controls Flowering Time and Seed Size in Rice and *Arabidopsis*

**DOI:** 10.3390/ijms232113448

**Published:** 2022-11-03

**Authors:** Shan Lu, Ning Zhang, Yazhen Xu, Hao Chen, Jie Huang, Baohong Zou

**Affiliations:** 1The State Key Laboratory of Crop Genetics and Germplasm Enhancement, Nanjing Agricultural University, Nanjing 210095, China; 2State Key Laboratory of Rice Biology, Institute of Nuclear Agricultural Sciences, Zhejiang University, Hangzhou 310029, China

**Keywords:** rice, heading date, florigen, MOS1, gene expression

## Abstract

The heading date and grain size are two essential traits affecting rice yield. Here, we found that *OsMOS1* promotes rice heading and affects its grain size. Knocking out *OsMOS1* delayed heading, while the overexpression of *OsMOS1* promoted heading in rice under long-day conditions. The transcriptions of the heading activators *Ehd1*, *Hd3a*, and *RFT1* were decreased and the heading repressor *Hd1* was increased in the *osmos1* mutant. Conversely, the overexpression of *OsMOS1* promoted the expressions of *Ehd1*, *Hd3a*, and *RFT1*, but inhibited the expression of *Hd1*. This suggests that *OsMOS1* may control heading in rice by modulating the transcriptions of *Ehd1*, *Hd3a*, *RFT1*, and *Hd1*. In addition, knocking out *OsMOS1* led to larger grains with longer grain lengths and higher grain weights. The seed cell size measurement showed that the cell lengths and cell widths of the outer glume epidermal cells of the *osmos1* mutant were greater than those of the wild type. Furthermore, we also found that the overexpression of *OsMOS1* in the *Arabidopsis mos1* mutant background could suppress its phenotypes of late flowering and increased seed size. Thus, our study shows a conserved function of *MOS1* in rice and *Arabidopsis*, and these findings shed light on the heading and seed size regulation in rice and suggest that *OsMOS1* is a promising target for rice yield improvement.

## 1. Introduction

As one of the most important cereal crops, rice (*Oryza sativa* L.) provides stable food for more than half the world’s population [1]. The heading date (Flowering time) and grain size are two major agronomic traits that affect yield in rice [2]. Early heading during the growing season without enough vegetative growth stage may lead to insufficient nutrient accumulation in rice [3]. Additionally, if the heading is too late during the growing season, the later reproductive growth will be affected by lower temperatures and lead to a decreased final yield in rice [4]. The grain size includes the grain length, width, and thickness, which are associated with grain weight [2]. In general, the final grain size of rice is coordinately regulated by cell proliferation and cell expansion in the spikelet hull [5]. Increasing the grain yield has become an urgent necessity in rice breeding and it is worthwhile to explore new genes that control grain size and heading date in rice for yield improvement.

The rice heading date is controlled by multiple factors, among which the photoperiod is a major factor [6]. Photoperiodic flowering consists of a complicated network that converges into the generation of a mobile flowering signal called florigen [7]. Florigen is synthesized in the leaves and then moved to the shoot apex to induce flowering [7]. As a short-day (SD) plant, the rice heading date is drastically promoted under SD conditions and inhibited under long-day (LD) conditions [8]. There are two florigen genes, namely *HEADING DATE 3a* (*Hd3a*) and *RICE FLOWERING LOCUS T1* (*RFT1*) in rice, which function as florigen in SD and LD, respectively [9,10]. The transcriptions of *Hd3a* and *RFT1* are mainly controlled by a B-type response regulator *EARLY HEADING DATE 1* (*Ehd1*) [11] and *Hd1*, which is an ortholog of the *Arabidopsis* floral activator *CONSTANS* (*CO*) [12,13]. *Hd1* promotes floral transition under SDs by up-regulating *Hd3a* and *RFT1* expression, while it strongly inhibits *Hd3a* and *RFT1* expression by directly repressing *Ehd1* transcription to restrain floral transition under LDs [14,15,16]. *Hd1* is controlled by *GIGANTEA* (*GI*), which inhibits flowering in rice by repressing the expression of *Hd3a* [11,12,17]. The floral signal transduction cascade mediated by GI-Hd1-Hd3a, in which *Hd1* receives signals from *GI* and then affects the expression of *Hd3a* to regulate the flowering time, has been found to be conserved between *Arabidopsis* and rice [15]. *Ehd1* integrates different upstream signals and promotes the expressions of *Hd3a* and *RFT1* under both LD and SD conditions [8,15]. The expression of *Ehd1* is up-regulated by several upstream positive regulators, including *Ehd2*, *Ehd3*, *Ehd4* [18,19,20], *Hd17* [21], *OsMADS50*, and *OsMADS51* [22,23]. Additionally, several negative regulators, including *Ghd7*, *Ghd 8* [24,25], and *DELAYED HEADING DATE1* (*Dhd1*) [26], prevent *Ehd1* expression under non-inductive photoperiods to repress heading. 

An evolutionarily conserved BAT2 (HLA-B ASSOCIATED TRANSCRIPT2) domain containing protein MODIFIER OF *snc1* (MOS1) was first identified as a suppressor of the autoimmune mutants *snc1* and *bonzai1* (*bon1*) in *Arabidopsis* [27,28]. Autoimmunity in *bon1* and *snc1* is caused by the constitutive activation of an *NLR* gene *SUPPRESSOR OF npr1-1 and CONSTITUTIVE 1* (*SNC1*) [29,30]. MOS1 interacts with TEOSINTE BRANCHED 1, CYCLOIDEA, and PCF 15 (TCP15) transcription factors, and might regulate the expression of *SNC1* by directly binding to its promoter [31]. *MOS1* has also been found to regulate the flowering time and endoreduplication [27]. MOS1 interacts with the spindle assembly checkpoint (SAC) component MAD2 and promotes flowering and inhibits endopolyploidization, but this function could be antagonized by another SAC component, MAD1 [27]. MOS1 interacts with the zinc-finger-containing transcription factor SUPPRESSOR OF FRIGIDA 4 (SUF4) to inhibit the expression of the central flowering time gene *FLOWERING LOCUS C* (*FLC*) and thus promote flowering [27]. The defection of endopolyploidization in the *mos1* mutant can be suppressed by the mutations of *SUF4*, but not *FLC*, indicating that *MOS1* modulates the endopolyploidization and flowering time through different genetic pathways [27]. In addition, *MOS1* negatively regulates sugar responses and anthocyanin biosynthesis in *Arabidopsis*, possibly at the transcriptional level [32]. Thus, the existing finding indicates that *MOS1* functions as a key coordinator in the regulation of growth and biotic and abiotic stress responses in *Arabidopsis*.

The functions of *MOS1* in *Arabidopsis* have been studied; however, the roles of *MOS1* in other plants, such as rice, remain unknown. In this study, *OsMOS1*, a rice gene homologous to *Arabidopsis MOS1*, was characterized. We assessed how *OsMOS1* controlled heading and grain size by generating its knock-out mutant and overexpression transgenic plants. In addition, we investigated the functional conservation of *MOS1*, which controls flowering time and grain size in rice and *Arabidopsis*, by introducing *OsMOS1* into *atmos1* mutants. Here, we demonstrate a new function of *MOS1* proteins in seed size control, in addition to their known roles in the flowering time of plants.

## 2. Results

### 2.1. Isolation and Expression of MOS1 in Rice

To gain a better understanding of MOS1 proteins in plants using information from Arabidopsis MOS1 (At4g24680), we selected a total of 12 plant species for *MOS1* gene identification based on their evolutionary distance and availability of full-genome sequences. The genome databases of these organisms were searched with BLASTP (2 March 2019, NCBI, https://www.ncbi.nlm.nih.gov/) using the Arabidopsis AtMOS1 protein sequence. The full-length protein sequences of 17 candidate MOS1 proteins were identified from 12 plant species. There were one or two MOS1 proteins in each of the 12 selected plant species. A phylogenetic tree was constructed using the full-length protein sequences of the 17 candidate MOS1 proteins. Phylogenetic analysis revealed that 12 plant species had two clades of MOS1 (Figure 1a). The rice protein (accession numbers LOC_Os12g37860) was identified as a MOS1 protein in rice by confirming that it contained the conserved BAT2 domain (Figure 1b). We then named this gene *OsMOS1*. Sequence analyses revealed that OsMOS1 encodes a protein consisting of 1460 amino acids (Appendix A). The deduced amino sequence of OsMOS1 shared 60% similarity with AtMOS1 (Appendix A). The protein domain analyzed by the SMART website tool (8 March 2019, http://smart.embl-heidelberg.de/) showed that OsMOS1 contained a conserved BAT2 domain at the N-terminal region. In addition to the BAT2 domain, OsMOS1 also contained a PRK10263 (DNA Translocase FtsK) domain, which was different from other MOS1 proteins, including AtMOS1 in plants (Figure 1b). 

To gain insights into the possible functions of the *OsMOS1* gene in rice, the expression patterns of the *OsMOS1* gene in different rice tissues were examined by quantitative reverse transcription PCR (qPCR). The qPCR results revealed that the *OsMOS1* gene was expressed over a wide range of tissues in the wild-type rice cultivar Nipponbare (NIP) (Figure 2). *OsMOS1* transcripts were detectable in all of the selected organs, including the root (three-leaf stage), stem (three-leaf stage), leaf (three-leaf stage), and panicle (young panicles at P2 stages) of the wild-type rice cultivar NIP and with the highest transcript levels in the young panicles (Figure 2). The expression patterns of the *AtMOS1* gene were also examined by taking advantage of the online, publicly available microarray data. The tissue expression patterns analyses using the BAR website (15 October 2022, http://www.bar.utoronto.ca/) revealed that the *AtMOS1* gene was also expressed over a wide range of tissues in the wild-type Col-0 (Appendix A). *AtMOS1* transcripts were detectable in all of the selected organs, including the roots (four-rosette leaf stage), stems (four-rosette leaf stage, second internode), leaves (rosette leaf 2), and siliques (stage 8 siliques), with the highest transcript levels in the siliques (Appendix A). These results reveal that *OsMOS1* and *AtMOS1* had similar tissue expression patterns, suggesting the functional conservation of MOS1 in rice and *Arabidopsis*.

### 2.2. OsMOS1 Promotes Heading Date in Rice

For the functional characterization of *OsMOS1*, we generated its loss of function mutants in the NIP or R7954 (*O. sativa L*. ssp. *indica*) background using CRISPR/Cas9 gene-editing technology [33]. Homozygous mutants were identified in the T_0_ generation by sequencing the PCR product containing the CRISPR/Cas9 target sites. Four *OsMOS1* mutants (*osmos1-1*, *osmos1-2*, *osmos1-3*, and *osmos1-4*) were obtained, and all but *osmos1-1* and *osmos1-2* were predicted to be loss-of-function (LOF) mutants (Figure 3). The *osmos1-1* mutant contained a three-base-pair (bp) deletion at the position of 468–470 (relative to the translation initiation site), leading to a 1-amino acid deletion (amino acid 75) in its predicted coding protein. The *osmos1-2* mutant contained a 6 bp deletion at the position of 468–473, leading to a 2-amino acid deletion (amino acids 75–76) in its predicted coding protein. The *osmos1-3* mutant contained a 13 bp deletion at the position of 457–469, leading to reading frame shifts in *OsMOS1* and a stop codon at amino acid 73. The *osmos1-4* mutant was generated by targeting sequences different from those of *osmos1-1*, *osmos1-2*, and *osmos1-3*, so they were independent from *osmos1-1*, *osmos1-2*, and *osmos1-3* and unlikely to have had the same off-target mutations, if any. The *osmos1-4* mutant contained a 58 bp deletion at positions 495–552, leading to reading frame shifts in *OsMOS1* and a stop codon at amino acid 123.

Heading date analysis found that *osmos1-3*, but not *osmos1-1* or *osmos1-2*, headed approximately 11 days (d) later than the wild-type (WT) NIP under natural long-day (NLD) (Nanjing, China) conditions (Figure 4a,b). Therefore, the *osmos1-1* and *osmos1-2* mutants were either weak reductions of functional mutants or had no compromised function in *OsMOS1*. Similar to *osmos1-3*, *osmos1-4* exhibited a significantly late heading phenotype compared with the control plant (Appendix A). 

To further evaluate the function of *OsMOS1* in the heading date, an *OsMOS1* overexpression vector was constructed and transformed into NIP. After the confirmation of the presence of the transgene by the PCR, we obtained 21 transgenic plants. The transcript levels of *OsMOS1* were significantly increased in the overexpressing transgenic lines, and three representative *OsMOS1*-overexpressing transgenic lines, i.e., #17, #18, and #21, were used for further analyses (Figure 4d). Compared with the WT, the *OsMOS1*-overexpressing plants headed approximately 7 d earlier under NLD conditions (Figure 4c,e).

### 2.3. OsMOS1 Regulates the Expression of the Heading Date Genes in Rice

To assess the molecular mechanisms by which *OsMOS1* regulates the heading date in rice, we examined the transcription levels of floral regulator genes in NIP, *osmos1-3*, and *OsMOS1*-overexpressing plants under LD growth conditions in a greenhouse. Florigen genes *Hd3a* and *RFT1*, heading repressor gene *Hd1*, and heading enhancer gene *Ehd1* were selected for further gene expression analysis. Considering the diurnal rhythmic expression patterns of the above floral regulator genes, the expression of *Hd1* was monitored at zeitgeber time (ZT) 14 h and *Ehd1*, *Hd3a*, and *RFT1* were monitored at ZT 2 h, as previously reported [17]. Our results showed that the transcripts of the two florigen genes *Hd3a* and *RFT1*, and the LD heading enhancer *Ehd1* were less abundant in the *osmos1-3* (Figure 5a–c), but more abundant in *OsMOS1*-overexpressing plants (Figure 5e–g). The transcript levels of the LD heading repressor *Hd1* were significantly increased in the *osmos1-3* mutant (Figure 5d), but reduced in the *OsMOS1*-overexpressing plants (Figure 5h). Taken together, these results suggest that *OsMOS1* may regulate heading by modulating the transcriptions of *Hd1*, *Ehd1*, *Hd3a*, and *RFT1* in rice under LD conditions.

### 2.4. MOS1 Negatively Regulates Seed Size in Rice and Arabidopsis

Grain phenotypic analysis indicated that *OsMOS1* also affected grain size. The grain size of *osmos1-3* was significantly larger than that of the wild-type NIP (Figure 6). The grain lengths, but not widths, of *osmos1-3* increased by 8.3% compared with those of NIP (Figure 6a–c). As a result, the *osmos1-3* mutant produced heavier grains (+8.2% in 1000-grain weight) than NIP (Figure 6d). However, there was no significant difference in the grain size between the wild-type NIP and *OsMOS1*-overexpressing plants (Appendix A). To investigate whether the difference between *osmos1-3* and NIP in grain size was caused by seed cell expansion, the outer glume epidermal cells were observed using a scanning electron microscope. As shown in Figure 6, the cell lengths and cell widths of *osmos1-3* were greater than those of NIP (Figure 6e,f). Taken together, these results show that *OsMOS1* negatively regulated grain size mainly by controlling the seed cell expansion.

### 2.5. Overexpression of OsMOS1 Affects the Late Flowering and Large Seed Size Phenotypes of the Arabidopsis mos1 Mutant

In *Arabidopsis*, *MOS1* functions as a positive regulator of the flowering time [27]. To investigate the functional conservation in the flowering time between rice and *Arabidopsis MOS1* homologs, we introduced the Ubi:*OsMOS1* transgene into the *atmos1-6* mutant. The *atmos1-6* mutant was a LOF mutant of *AtMOS1* with an 80 bp deletion in the fourth exon of *AtMOS1* [27]. After the confirmation of the presence of the transgene by PCR, we obtained five independent transgenic plants and RT-PCR analysis showed that *OsMOS1* was successfully expressed in the transgenic *Arabidopsis* lines (Appendix A). Two representative transgenic lines, #1 and #2, were used for further analyses. The *atmos1-6* mutant exhibited a late-flowering phenotype, whereas the Ubi:*OsMOS1*/*atmos1-6* transgenic lines restored the late-flowering phenotype of *atmos1-6* to the wild type (Figure 7a–c). As AtMOS1 promoted flowering by inhibiting the expression of the *FLC* gene [27], we later checked the expression of the *FLC* gene in Ubi:*OsMOS1*/*atmos1-6* transgenic plants under LD conditions. We found that the upregulation of the *FLC* gene in the *atmos1-6* mutant was restored to the WT level in Ubi:*OsMOS1*/*atmos1-6* transgenic plants (Figure 7d). These results suggest that the overexpression of *OsMOS1* rescued the late flowering phenotype of the *atmos1-6* mutant by down-regulating the expression of *FLC*.

As with the *osmos1-3* mutant, we found that the seed size of the *atmos1-6* mutant was significantly larger than that of the wild-type Col-0 (Figure 8a). Compared with Col-0, the seed lengths and widths of *atmos1* mutants increased by 26.3% and 20.6%, respectively. Thus, these results suggest that *MOS1* has a conserved function in the plant seed size. Furthermore, we found that the overexpression of *OsMOS1* also affected the seed size in *Arabidopsis*. The seed lengths of the two Ubi:*OsMOS1*/*atmos1-6* transgenic lines were 488.76 μm and 472.93 μm respectively, which were significantly smaller than those of the *atmos1-6* (532.15 μm) mutant but still larger than those of the WT (421.46 μm) (Figure 8b). In addition, the grain widths of the two Ubi:*OsMOS1*/*atmos1-6* transgenic lines were 279.06 μm and 274.15 μm, respectively, which were significantly smaller than those of the *atmos1-6* mutant (310.63 μm), but still larger than those of the WT (257.56 μm) (Figure 8c). Thus, these results suggest that the overexpression of *OsMOS1* partially rescued the seed size phenotype of the *Arabidopsis mos1-6* mutant.

## 3. Discussion

*MOS1* performs various regulatory activities in multiple aspects of plant growth, including the flowering time, cell cycle, and stress responses in *Arabidopsis* [27,28,31,32]. Here, we focused on the functional characterization of a rice *MOS1*, *OsMOS1*, which is a close homolog of *Arabidopsis MOS1* as it contains the conserved BAT2 domain. Our results show that *OsMOS1* is involved in the regulation of the heading date and seed size in rice. It controls rice heading by regulating the expressions of several floral-regulation genes and seed size in both rice and Arabidopsis.

*OsMOS1* regulates both the heading date and grain size in rice. It is possible that the primary function of *OsMOS1* is to promote the heading date in rice. As a longer vegetative growth period usually results in more accumulation of nutrients, the larger grain size in the *osmos1-3* mutant was likely to be promoted by the increased accumulation of nutrients. Furthermore, we found that the role of *MOS1* in regulating the flowering time and seed size appeared to be largely conserved between rice and *Arabidopsis*. The heterologous overexpression of *OsMOS1* in the *Arabidopsis atmos1* mutant could rescue or partially rescue its late flowering and large seed phenotypes. Meanwhile, it is worth noting that the loss of function of *OsMOS1* or *AtMOS1* will increase the seed size, but *OsMOS1* cannot fully rescue the seed size phenotype of the *atmos1* mutant, suggesting that there are other decisive factors involved. Therefore, despite the functional conservation in seed size between *OsMOS1* and *AtMOS1*, the specific regulation mechanism may still have some differences.

There are two photoperiod heading pathways that are mediated by GI-Hd1-Hd3a/RFT and GI-Ehd1-Hd3a/RFT in rice. Here, we found that the expressions of *Ehd1* and both the florigens, *RFT1* and *Hd3a*, were down-regulated in the *osmos1-3* mutant but up-regulated in the *OsMOS1*-overexpressing transgenic lines under LD conditions. While the expression of *Hd1*, a repressor of heading under LD conditions, was up-regulated in the *osmos1-3* mutant, it was down-regulated in *OsMOS1*-overexpressing transgenic lines. Taken together, our results suggest that *OsMOS1* may regulate heading through both the Hd1- and Ehd1-mediated pathways in rice. Hd1 can also inhibit the heading date by directly repressing *Ehd1* transcription to indirectly regulate the expressions of *Hd3a* and *RFT1* under LD conditions [34]. Thus, this raises the possibility of the coordinated functioning of *OsMOS1* in the above two heading regulation pathways in rice, which needs further exploration.

In the *osmos1-3* mutant, the expressions of the two florigens *RFT1* and *Hd3a* were significantly repressed, leading to heading being delayed. The expression of the florigen *AtFT*, the homolog of *RFT1* in *Arabidopsis*, was also down-regulated in the *atmos1* mutant [27]. In addition, the up-regulation of another flowering-related gene, *AtFLC*, in the *atmos1* mutant could be recovered by *OsMOS1* overexpression in Arabidopsis. Thus, these results indicate that MOS1 regulates the flowering time in rice and Arabidopsis by promoting florigen gene expression. Taken together, our study shows that *OsMOS1* has important functions in regulating the rice heading date and grain size. Consequently, it is a promising gene target for breeding new rice varieties with enhanced yield.

## 4. Materials and Methods

### 4.1. Plant Materials and Growth Conditions

The wild-type rice in this study were NIP and R7954. The seeds of WT, *osmos1* mutants, and transgenic plants were soaked in water at 28 °C in the dark for 3 d, then germinated for 2 d at 37 °C. Rice seedlings were planted in soil and grown under a 14/10 h light/dark cycle in a chamber at 28 °C or in a natural field during the summer in Nanjing, China. The seeds of *Arabidopsis* wild-type Col-0, *atmos1-6* mutant [27], and Ubi:*OsMOS1*/*atmos1-6* transgenic plants were sown in soil and grown in a chamber with 16 h light/8 h dark, 100 µmol/s/m^2^, and 60% humidity at 22 °C after stratifying at 4 °C for 4 d.

### 4.2. Generation of the CRISPR/Cas9 Mutants

To generate the CRISPR/Cas9 mutants, the specific guide RNA spacer sequences of the *OsMOS1* gene were selected using the CRISPR-PLANT website (20 March 2019, http://www.genome.arizona.edu/crispr/CRISPRsearch.html). Three guide RNA targets in the exon *OsMOS1* were cloned into the pHUE411 construct [33] and then introduced into the calli of the rice cultivar NIP or R7954 via EHA105-meditated methods [35]. The genomic DNA of Ubi:*OsMOS1* transgenic lines and the *osmos1* mutants was extracted, the genomic regions surrounding the CRISPR/Cas9 target sites for *OsMOS1* were amplified by PCR, and the segment was sequenced to screen for mutants.

### 4.3. Generation of Transgenic Plants

The full-length coding sequence (CDS) of *OsMOS1* was amplified from the total RNA of wild-type NIP by the reverse transcription (RT)-polymerase chain reaction (PCR) using gene-specific primer sets (Appendix A). The amplicons were digested with restriction enzymes BamHI and KpnI and the resulting restriction fragments were subcloned into pCUbi1390, a plant binary vector harboring the Ubi promoter.

For the generation of *OsMOS1* transgenic plants in rice, the Ubi:*OsMOS1* construct was transformed into *Agrobacterium tumefaciens* EHA105 and then introduced into the calli of the rice cultivar Nip by EHA105-meditated methods [35]. The transgenic plants were screened with a hygromycin-resistant culture medium and the overexpression of *OsMOS1* in independent T_2_ plants was confirmed by qPCR analysis. The heading date was defined as the number of days of seed-soaking until the appearance of the first (main) panicle. The heading date was determined by counting the heading date from 15 T_2_ generation plants.

For the generation of *OsMOS1* transgenic plants in *Arabidopsis*, the Ubi:*OsMOS1* recombinant construct was introduced into the *atmos1-6* mutant using the floral dip method [36]. The transgenic plants were screened with hygromycin-resistant culture medium and the overexpression of *OsMOS1* in independent T_2_ plants was confirmed by qPCR analysis. The flowering time was determined by counting the total number of rosette leaves from 15 T_2_ generation plants.

### 4.4. RNA Extraction and qPCR

The total RNA was extracted using TRIzol reagent (RP1001, Bioteke, Beijing, China) according to the manufacturer’s instructions. The complementary DNA (cDNA) was synthesized by using the Transcript 1st strand cDNA synthesis kit (R323-01, Vazyme, Nanjing, China). qPCR was performed by using an SYBR Green Supermix with gene-specific primers in a Bio-Rad CFX96 real-time PCR detection system (Bio-Rad Laboratories, Hercules, CA, USA). Rice *ACTIN* and *Arabidopsis UFP* were used as the internal control genes for normalization. The expression levels were calculated by the comparative cycle threshold method as previously reported [37].

For the floral regulator gene expression in rice, fully emerged leaf blades were sampled 45 days after germination (DAG). The expression of *Hd1* was observed at ZT 14 h; other floral regulator genes were monitored at ZT 2h. The transcript levels of *OsMOS1* in Ubi:*OsMOS1* transgenic rice were detected by qPCR in the 30-day leaves under LD (14 day/10 night) chamber growth conditions. The transcript levels of *OsMOS1* in Ubi:*OsMOS1*/*atmos1-6* transgenic *Arabidopsis* were detected by RT-PCR in the 14-day leaves under LD (14 day/10 night) growth conditions. The transcript levels of *AtFLC* were detected by qPCR in rosette leaves of 3-week-old plants under LD growth conditions. The primer sequences used for the PCR are listed in Appendix A.

### 4.5. Seed Traits Measurement

The seed size and 1000-grain weight were measured when the rice and Arabidopsis plants were completely matured. The grain length, grain width, and 1000-grain weight were measured after the grains had been harvested and treated at 37 °C for at least 5 days. For the rice grain length and width measurement, 50 grains were measured once and the measurement was repeated 3 times. For the Arabidopsis seed length and width measurement, 500 grains were measured once and the measurement was repeated 3 times. For rice 1000-grain weight measurement, 500 grains were weighed once and the measurement was repeated 3 times. The observation of the outer epidermal cells of NIP and *osmos1-3* mutant lemmas using a scanning electron microscope was performed as described previously [38]. The cell length and width were measured with the software Image J. The seed cell length and width were determined by averaging the values of three replicates (each replicate consisted of at least 30 cells).

## Figures and Tables

**Figure 1 ijms-23-13448-f001:**
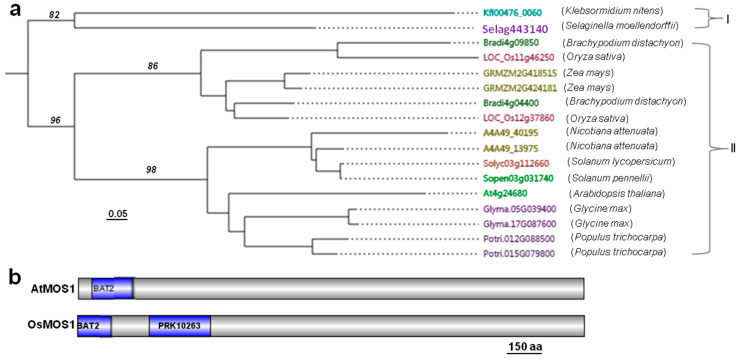
**Homology analysis of MOS1.** (**a**) Phylogenetic tree of MOS1 proteins from 12 plant species. The tree was constructed by using MEGA6.06 software based on the MOS1 protein sequences from the National Coalition Building Institute (2 March 2019, NCBI, https://www.ncbi.nlm.nih.gov/). The scale bar indicates the branch length. (**b**) Protein structures of AtMOS1 and OsMOS1. PRK10263 (DNA Translocase FtsK domain).

**Figure 2 ijms-23-13448-f002:**
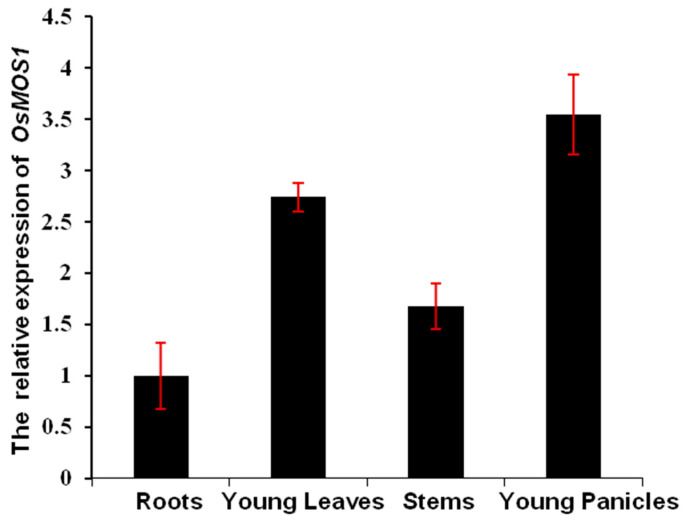
**Expression profiles of *OsMOS1*.** Relative expression levels of *OsMOS1* in different rice tissues of NIP checked by qPCR. The rice *ACTIN* gene was used as the internal control. Error bars indicate the standard deviation, *n* = 3.

**Figure 3 ijms-23-13448-f003:**
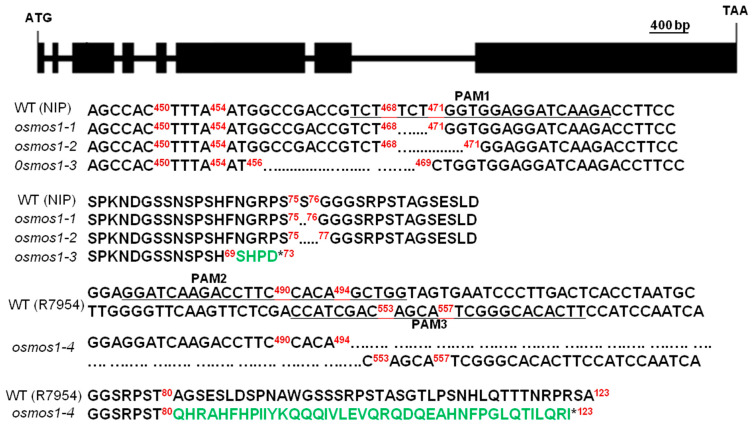
**Mutants of *OsMOS1* generated by CRISPR/Cas9 technology.** Diagram of the genomic region and mutations of *OsMOS1.* The top diagram is the gene structure of the *OsMOS1* genomic sequence. Exons and introns are indicated by black boxes and lines between boxes, respectively. The bottom panel shows the sequence alignments between the mutants and wild type with the genomic sequences above and the protein sequence below. The dotted line indicates the deletion of a base pair or amino acid sequence, green capital letters indicate amino acid missense mutations, * sign indicates a generated stop codon, and PAM indicates the protospacer adjacent motif.

**Figure 4 ijms-23-13448-f004:**
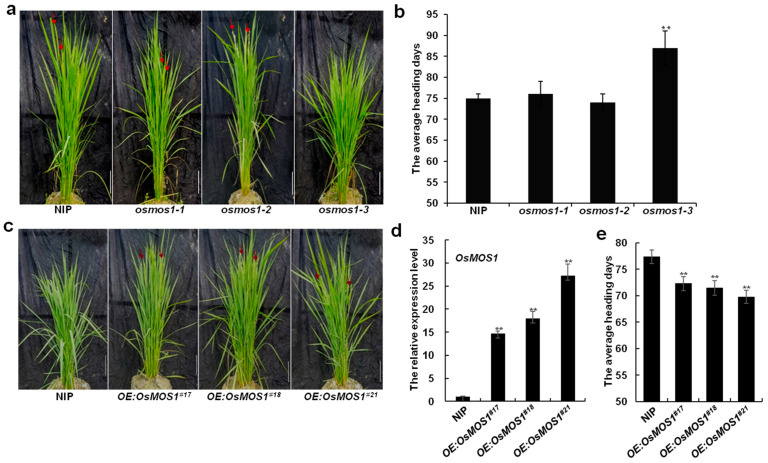
***OsMOS1* promotes rice heading date.** (**a**) Heading phenotype of Nip and *osmos1* mutants under NLD conditions. NIP, Nipponbare. Bars = 20 cm. Red arrows indicate panicles. (**b**) Statistical analysis for the heading date of Nip and *osmos1* mutants under NLD conditions. Error bars indicate the standard deviation, *n* = 15. Asterisks indicate a significant difference between NIP and *osmos1* mutants (** *p* < 0.01, Student’s *t*-test). (**c**) Early heading date phenotypes of *OsMOS1*-overexpressing lines under NLD conditions. The numbers 17#, 18#, and 21# represent three independent transgenic lines. The red arrows indicate panicles. Bars = 20 cm. (**d**) Relative transcript levels of *OsMOS1* in *OsMOS1* transgenic lines. The rice *ACTIN* gene was used as the internal control. Error bars indicate the standard deviation, *n* = 3. Asterisks indicate a significant difference between NIP and *OsMOS1*-overexpressing lines (** *p* < 0.01, Student’s *t*-test). (**e**) Statistical analysis for the heading date of Nip and *OsMOS1*-overexpressing lines under NLD conditions. Error bars indicate the standard deviation, *n* = 15. Asterisks indicate a significant difference between NIP and *OsMOS1*-overexpressing lines (** *p* < 0.01, Student’s *t*-test).

**Figure 5 ijms-23-13448-f005:**
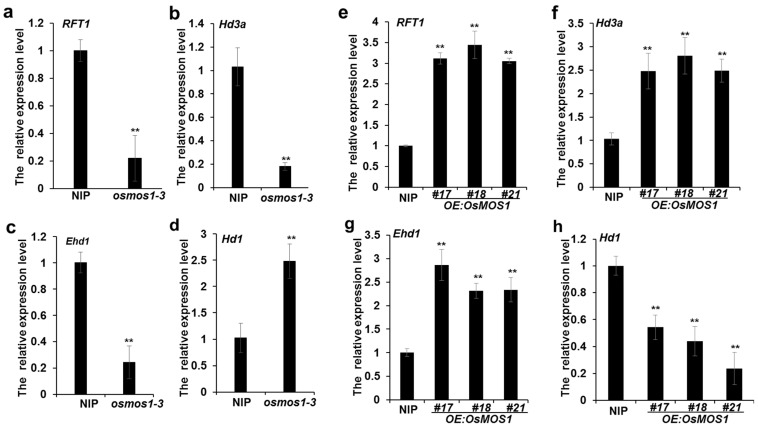
Expressions of the floral regulator genes in Nip, *osmos1-3*, and *OsMOS1*-overexpressing lines under LD conditions. (**a**–**d**) Expressions of the *RFT1* (**a**), *Hd3a* (**b**), *Ehd1* (**c**), and *Hd1* (**d**) in Nip and *osmos1-3* under LD conditions. (**e**–**h**) Expressions of *RFT1* (**e**), *Hd3a* (**f**), *Ehd1* (**g**), and *Hd1* (**h**) in NIP and *OsMOS1*-overexpressing lines under LD conditions. The fully emerged leaf blades from Nip, *osmos1-3*, and *OsMOS1*-overexpressing plants under LD growth conditions were sampled at 45 days after germination. The expression of *Hd1* was observed at ZT 14 h and other floral regulator genes were monitored at ZT 2 h. The rice *ACTIN* gene was used as the internal control. Error bars indicate the standard deviation, *n* = 3. Asterisks indicate a significant difference between NIP and *osmos1-3* (** *p* < 0.01, Student’s *t*-test).

**Figure 6 ijms-23-13448-f006:**
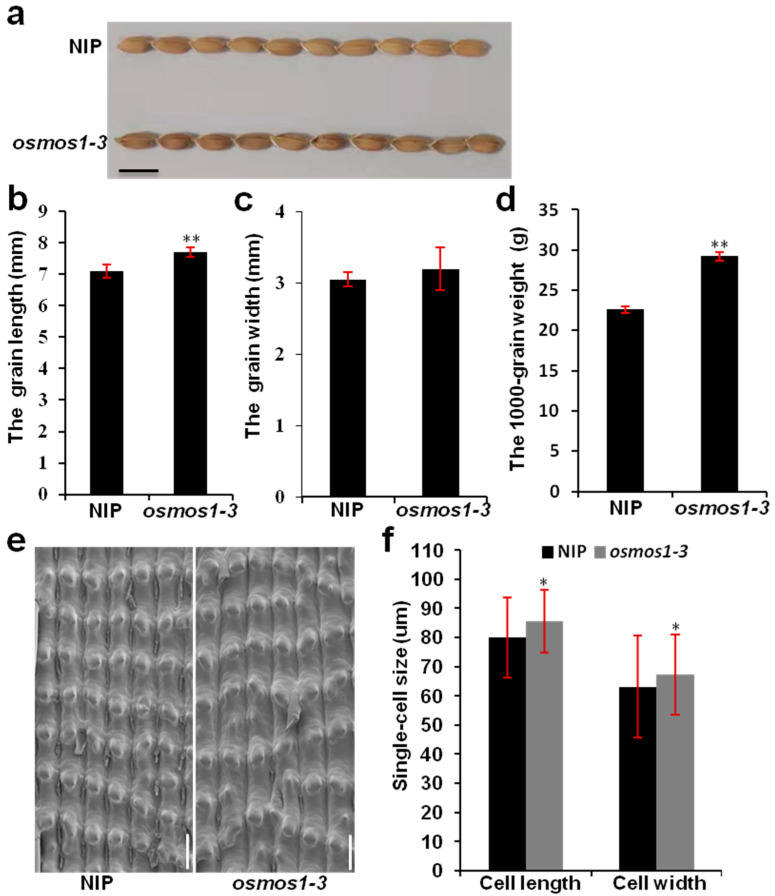
Analyses of the grain and cell sizes of NIP and the *osmos1-3* mutant. (**a**) Grain morphology of NIP and the *osmos1-3* mutant. Scale bars correspond to 1 cm. (**b**,**c**) Statistical analyses of the grain length (**b**) and grain width (**c**) between NIP and the *osmos1-3* mutant. Error bars indicate the standard deviation, *n* = 3 × 50. Asterisks indicate a significant difference compared with NIP (** *p* < 0.05, Student’s *t* test). (**d**) Statistical analysis of the 1000-grain weight between NIP and the *osmos1-3* mutant. Error bars indicate the standard deviation, *n* = 3 × 500. Asterisks indicate a significant difference compared with NIP (** *p* < 0.01, Student’s *t* test). (**e**) Electron microscopy scan of the outer glume surface of a mature grain from NIP and *osmos1-3* mutants. Scale bars correspond to 65 μm. (**f**) Quantification of a single cell’s length and width per mm^2^ in (**d**). Asterisks indicate a significant difference compared with NIP (* *p* < 0.05, Student’s *t* test).

**Figure 7 ijms-23-13448-f007:**
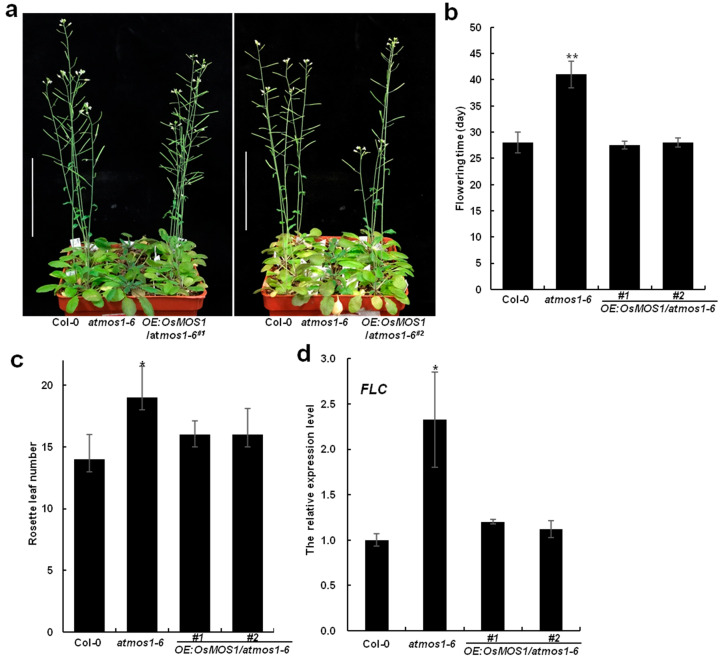
**Analyses of flowering time and *FLC* gene expression in *OsMOS1*-overexpressing transgenic plants in *Arabidopsis*.** (**a**) Flowering phenotypes of Col-0, *atmos1-6*, *OE:OsMOS1/atmos1-6 #1*, and *OE:OsMOS1/atmos1-6 #2* plants under LD conditions. Scale bars = 5 cm. (**b**) Statistical analysis for flowering time of Col-0, *atmos1-6*, *OE:OsMOS1/atmos1-6 #1*, and *OE:OsMOS1/atmos1-6 #2* plants under LD conditions. Asterisks indicate a significant difference compared with Col-0 (** *p* < 0.01, Student’s *t*-test). Error bars indicate the standard deviation, *n* = 15. (**c**) Statistical analysis for the rosette leaf numbers at the bolting stage of Col-0, *atmos1-6*, *OE:OsMOS1/atmos1-6 #1*, and *OE:OsMOS1/atmos1-6 #2* plants under LD conditions. Asterisks indicate a significant difference compared with Col-0 (* *p* < 0.05, Student’s *t* test). Error bars indicate the standard deviation, *n* = 15. (**d**) Expression of the *FLC* in Col-0, *atmos1-6*, *OE:OsMOS1/atmos1-6 #1*, and *OE:OsMOS1/atmos1-6 #2* plants under LD conditions. The *Arabidopsis UFP* gene was used as the internal control. Error bars indicate the standard deviation, *n* = 3. Asterisks indicate a significant difference compared with Col-0 (* *p* < 0.05, Student’s *t* test).

**Figure 8 ijms-23-13448-f008:**
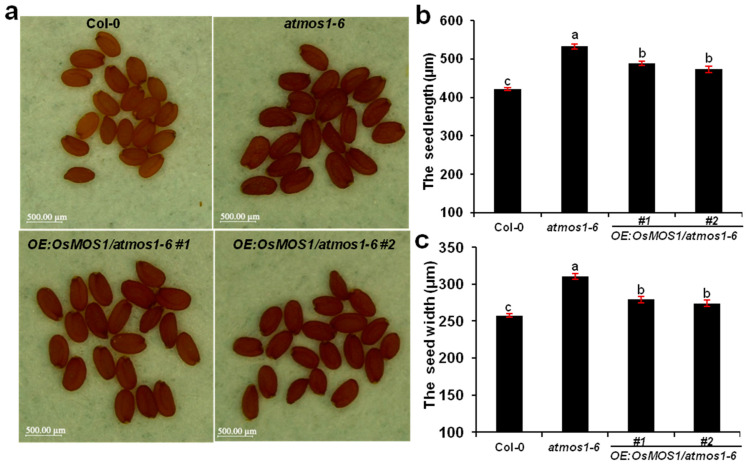
**Analyses of seed size in Col-0, *atmos1-6*, and *OE:OsMOS1/atmos1-6* transgenic plants.** (**a**) Seed morphology of Col-0, *atmos1-6*, and *OE:OsMOS1/atmos1-6* transgenic plants. Scale bars correspond to 500 μm. (**b**,**c**) Statistical analysis of seed length (**b**) and seed width (**c**) in Col-0, *atmos1-6*, and *OE:OsMOS1/atmos1-6* transgenic plants. Error bars indicate the standard deviation, *n* = 3 × 500. Different letters indicate significant differences based on Duncan’s multiple range test among the means via ANOVA (*p* < 0.05). The cell length and width were determined by averaging the results for three replicates (each replicate consisted of at least 30 cells).

## Data Availability

Genes from *Arabidopsis* in this study can be accessed on TAIR (2 March 2019, www.arabidopsis.org) under the following accession numbers: *AtMOS1* (AT4G24680), *AtUFP* (AT4G01000), and *AtFLC* (AT5G10140). Genes from rice in this study can be accessed on RGAP (2 March 2019, http://rice.uga.edu/) under the following accession numbers: *OsMOS1* (LOC_Os12g37860), *OsHd3a* (LOC_Os06g06320), *OsRFT1* (LOC_Os06g06300), *OsEhd1* (LOC_Os10g32600), and *OsHd1* (LOC_Os06g16370).

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
