# Peer review of "Functional Conservation and Divergence of MOS1 That Controls Flowering Time and Seed Size in Rice and Arabidopsis"

_ijms, 2022, doi:10.3390/ijms232113448_

Round 1

Reviewer 1 Report

The manuscript presented the functional conservation and diversity of MOS1 in controlling flowering time and seed size in rice and Arabidopsis. The authors analyzed the function of OsMOS1 in heading data and grain size in rice by knockout and overexpression experiments. They further investigated the functional conservation of MOS1 in rice and Arabidopsis by overexpressing OsMOS1 in atmos1 mutants. The following comments may improve the manuscript:

1.     Gene names should be italicized. Specie names in Latin also should be italicized. There are too many errors in the format of of gene and specie names.

2.     Please use the past tense to present the experimental results in abstract.

3.     Line 21-23, “The same as in rice, knocking out AtMOS1 in Arabidopsis also led to larger seed than wild type. Furthermore, overexpression of OsMOS1 suppresses the late flowering and increased seed size of the Arabidopsis mos1 mutant”. The two sentences need to be reworded.

4.     Line 31, “approximately” should be removed.

5.     Line 34, “Also if the heading is too late, the later reproductive growth will be affected by 34 lower temperatures and lead to decreased final yield in rice.” The sentence needs to be reworded.

6.     Line 35, 36: “final” should be removed.

7.     Line 38, Global population is growing, but not at a very fast rate right now. Global population growth rate has already fallen below 1% in 2020.

8.     Line 43, 44, “is synthesized”, “move”. Please use a consistent voice.

9.     Line 46, “encoding” should be removed.

10.  Line 52, It is inappropriate to claim that Hd1 directly repress Ehd1 transcription. There is no evidence that Hd1 protein could bind to the cis-regulatory region in Ehd1.

11.  Line 56-58, “Ehd2/3/4”, “OsMADS50/51” “Ghd7/8”. The different genes should not be abbreviated into one word.

12.  Line 64, As you mentioned, Hd1 can regulate the expression of Ehd1, so the two pathways are not independent.

13.  Line 94, It is preferable to provide the identification details of AtMOS1 homologs, such as phylogenic relationship and amino acid alignment.

14.  Details of figures, such as the position of arrows and the size of domain name, should be carefully checked.

15.  The singular and plural form in text should be checked. For example, Line 105, “Protein structure of AtMOS1 and OsMOS1”.

16.  Line 121, The sentence is inappropriate. The dynamic expression of OsMOS1 during panicle development didn’t suggest its function in regulating heading date, because the florigen is synthesized in the leaves and then transferred to the shoot apex to induce flowering.

17.  Figure 2, Please keep the number of decimal places of numbers uniform, and keep the singular and plural of nouns uniform.

18.  Figure 3, There are two PAMs in figure. Why?

19.  Line 152-161, Among the osmos1-1, osmos1-2 and osmos1-3, only osmos1-3 was null mutant. It is preferable to place osmos1-4 in the main body rather than in the supplementary materials.

20.   Line 152-161, Characterization of photoperiodic response of flowering time genes is important for their usage in breeding practices and for deciphering regulatory network of rice flowering time. It is preferable to test the mutants under both NSD and NLD conditions.

21.  Line 156, osmos1-4 might be a non-functional allele, but not a strong allele.

22.  Figure 4, Please modify “The average heading date (Day)” into “Heading date (d)”.

23.  Line 191, Why did you use “further” in this sentence.

24.  Line 220, The cell width was greater in the mutant than WT, while no significant differences was observed for grain width between the two materials. Why?

25.  Figure 6, Please divide the figure 6b into two figures. What is the meaning of “n=50” and “n=5”? 50 grains or 50 plants? This determined the accuracy of phenotypic measurements and the reliability of conclusions of manuscript. Please provide the details of phenotyping in MM.

26.  Line 269, The comma is not in English format.

27.  Figure 8, Please divide the figure 8b into two figures, because the Duncan’s multiple test was performed separately for seed length and width.

28.  Line 297-299, “It is possible that the primary function of OsMOS1 is to positively regulate heading date in rice. The larger grain size in osmos1 mutants is promoted by later heading. As the longer vegetative growth period usually results in a more accumulation of nutrients.” This speculation was unfounded. There is no necessary correlation between longer heading date and larger grain size, either in breeding practice or in gene characterization.

29.  Line 331-338. “these results indicate that MOS1 regulates flowering time in rice and Arabidopsis through similar regulatory mechanisms by promoting florigen gene expression”. This conclusion is contradictory to the results. The results showed that OsMOS1 regulated heading data partially through Ehd1-Hd3a/RFT1, the unique pathway in rice. Most flowering time genes regulate flowering time by affecting expressions of florigen genes. We cann’t claim that regulatory mechanisms of these genes are all similar.

Author Response

Thank you very much for your valuable comments on our MS, which benefited us a lot and made our article more complete and refined. Please see the attachment for our detail response to your comments.

Reviewer 2 Report

The authors revealed a conserved function of MOS1 on heading and seed size in rice and Arabidopsis. The results are interesting and OsMOS1 might be a promising target for rice yield improvement. Several comments as follows.

1.     For the homology study of MOS1, it would be better to do a phylogenetic tree of MOS1 in different plants.

2.     Knocking out OsMOS1 in rice leads to larger grain. But the seed size phenotype was missed in this article. It would be interesting to check the seed size phenotype of the OsMOS1 overexpression plants as well.

3.     The bars lacked in Figure 1a. Please add the explanation of arrows in Figure 1.

Author Response

(The authors gave the same response as above.)

Reviewer 3 Report

Rice is one of the most important cereal crops in the world by serving as a staple food for more than half of the world population. The grain yield of rice is determined by several factors, including heading date and grain size. Here, the authors identified one new gene OsMOS1 which regulates both heading and grain size in rice. They found OsMOS1 positive regulate rice heading by modulating the expression of Ehd1, Hd1, Hd3a and RFT1 expression. The authors also provide evidence that OsMOS1 regulates grain size by controlling seed cell expansion. Finally, they proved MOS1 has a conserved role in flowering and seed size regulation in Arabidopsis and rice. The article is overall well written and clear. I just feel some of the essential data should be provided and some written issues should be corrected.

Additional major issues:

1.     the authors should do a sequence alignment to compare the protein similarity of AtMOS1 and OsMOS1.

2.     Knocking out OsMOS1 leads to larger grain and its heterologous overexpression in Arabidopsis suppresses the large seed size phenotype of atmos1 mutant. But what about the OsMOS1 overexpression in rice? The seed size phenotype of OsMOS1 overexpression transgenic plant in rice was missed in the MS for unknown reason.

3.     Some references were missed in the MS, such as line 63-64.

Minor issues

1.       The authors need to revise the title of the paper, the current one has grammatical errors. Considering that change “control” to “controlling” or some other ways.

2.       Pay attention to the tense in the Abstract. There are even two different tenses (“suppresses” and “increased”) in one sentence of Line 26-27.

3.       Keywords are present in the title, choose others

4.       Pay attention to the use of singular and plural. Line 17 “transcription” (also in Line 51) should be “transcriptions”, and the following “was” should be “are”. Line 189, should be “patterns”? Please have a check overall

5.       Line 39- Lead should leads

6.       Line 40-41. Considering “Grain size includes grain length, width, and thickness which is associated with grain weight”.

7.       Line 57- ‘the Hd3a and RFT1’ should be ‘Hd3a and RFT1’

8.       Line 119- ‘Next, qPCR’ should be ‘next we conducted quantitative reverse transcription PCR (qPCR )’

9.       Line 139, considering “The osmos1-1 mutant contained a 3-base pair (bp) deletion at the positions of 468-470 (relative to the translation initiation site), leading to a 1-amino acid deletion (amino acid 75) in its predicted coding protein”. Make the similar modification in the following two sentences and the next paragraph.

10.  Line 146, delete “regions”.

11.  Line 184- ‘To further assess the molecular mechanisms of OsMOS1’ should be ‘To assess the molecular mechanisms of OsMOS1’

12.  Line 293- OsMOS1 also has multiple functions in regulating should be OsMOS1 has multiple functions in regulating

13.  Line 317- RFT1 and Hd3a was were down-regulated in osmos1 should be RFT1 and Hd3a were down-regulated in osmos1

14.  Line 336- The wild-type rice in this study was the japonica rice variety Nipponbare (NIP) should be The wild-type rice in this study was the japonica rice variety NIP

15.  Line 359- by quantitative reverse transcription PCR (qPCR) analysis should be by qPCR

16.  Add bars in figure 4a. Line 170, change “osmos1” to “osmos1-3”. There is a “;” in line 172, but sometimes you use “,” in the case, please unify these. And also unify the “;” in line 173

17.  In figure 2c, why the “Why the abscissa of the last three columns are italics? Change it

18.  In the figure legend of fig2, pay attention of the use of “the”. “The expression of MOS1” should delete “The”. Please have a check overall. The legend of fig2c should add the method. Line 158, delete “the”

19.  Add bars in figure 4a. Line 170, change “osmos1” to “osmos1-3”. There is a “;” in line 172, but sometimes you use “,” in the case, please unify these. And also unify the “;” in line 173

20.  The figure legend for Figure 6a is not right

21.  In figure 7, the font of the ordinate seems different from that of other pictures, please unify the font and size of words in all the figures. Significance analysis should be done in fig 7a.

22.  Line 286, Figure 8 Analyses of seed size in Col-0,atmos1-6, OE:OsMOS1 transgenic plants should be Figure 8. Analyses of seed size in Col-0, atmos1-6 and OE:OsMOS1 transgenic plants”.

Author Response

(The authors gave the same response as above.)

Author Response

(The authors gave the same response as above.)

Round 2

Reviewer 1 Report

This version of the manuscript is a significant improvement over the previous one. I have only one suggestion for consideration of the authors. Please present how many plants the authors used for phenotype measurement in the experiments.

Author Response

Thanks for your comments which benifit us a lot. We have add the number of plants used for the phenotyping to the MS.

Reviewer 4 Report

Attached in PDF

Author Response

Thanks for your usefull comments which benifit us a lot. For the response to your comments, please see the attachment.
